Proceedings of the 6th Symposium on Advances in Approximate Bayesian Inference, 2024 1–16

# Fluctuation without dissipation: Microcanonical Langevin Monte Carlo

**Jakob Robnik**                                                    JAKOB_ROBNIK@BERKELEY.EDU
*Department of Physics, University of California, Berkeley, CA 94720, USA*

**Uroš Seljak**                                                      USELJAK@BERKELEY.EDU
*Department of Physics, University of California, Berkeley, CA 94720, USA*
*Lawrence Berkeley National Laboratory, 1 Cyclotron Road, Berkeley, CA 93720, USA*

## Abstract

Stochastic sampling algorithms such as Langevin Monte Carlo are inspired by physical systems in a heat bath. Their equilibrium distribution is the canonical ensemble given by a prescribed target distribution, so they must balance fluctuation and dissipation as dictated by the fluctuation-dissipation theorem. We show that the fluctuation-dissipation theorem is not required because only the configuration space distribution, and not the full phase space distribution, needs to be canonical. We propose a continuous-time Microcanonical Langevin Monte Carlo (MCLMC) as a dissipation-free system of stochastic differential equations (SDE). We derive the corresponding Fokker-Planck equation and show that the stationary distribution is the microcanonical ensemble with the desired canonical distribution on configuration space. We prove that MCLMC is ergodic for any nonzero amount of stochasticity, and for smooth, convex potentials, the expectation values converge exponentially fast. Furthermore, the deterministic drift and the stochastic diffusion separately preserve the stationary distribution. This uncommon property is attractive for practical implementations as it implies that the drift-diffusion discretization schemes are bias-free, so the only source of bias is the discretization of the deterministic dynamics. We apply MCLMC to a $\phi^4$ model on a 2d lattice, where Hamiltonian Monte Carlo (HMC) is currently the state-of-the-art integrator. MCLMC converges 12 to 32 times faster than HMC on an $8 \times 8$ to $64 \times 64$ lattice, and we expect even higher improvements for larger lattice sizes, such as in large scale lattice quantum chromodynamics.

## 1. Introduction

Sampling from a known probability distribution $e^{-S(\boldsymbol{x})}/Z$ with a possibly unknown normalization constant $Z$ is an important problem in many scientific disciplines, ranging from Bayesian statistics to statistical physics and quantum field theory. If $\boldsymbol{x}$ is high-dimensional, methods that use the gradient $\nabla S(\boldsymbol{x})$ are vastly more efficient than gradient-free MCMC, such as random walk Metropolis-Hastings (Metropolis et al., 2004).

Golden standard gradient-based methods are Hamiltonian Monte Carlo (HMC) (Duane et al., 1987) and (underdamped) Langevin Monte Carlo (LMC) (see e.g. (Leimkuhler and Matthews, 2015)). Both are based on the physics of a particle with position $\boldsymbol{x}(t)$, momentum $\boldsymbol{\Pi}(t)$, moving in an external potential $S(\boldsymbol{x})$. The dynamics is encoded in the Hamiltonian function $H(\boldsymbol{x}, \boldsymbol{\Pi}) = \frac{1}{2}|\boldsymbol{\Pi}|^2 + S(\boldsymbol{x})$ which gives rise to the Hamiltonian equations, a deterministic system of ordinary differential equations (ODE) for the phase space variables $\boldsymbol{z} = (\boldsymbol{x}, \boldsymbol{\Pi})$. HMC adds on top of that the occasional momentum resampling. Langevin

dynamics on the other hand, additionally models microscopic collisions with a heat bath by introducing damping and the diffusion process, giving rise to a set of Stochastic Differential Equations (SDE). Damping and diffusion are tied together by the fluctuation-dissipation theorem, ensuring that the probability distribution $\rho_t(\boldsymbol{z})$ of finding the particle at location $\boldsymbol{z}$ in the phase space converges to $\exp[-H(\boldsymbol{z})]$, which is known as the canonical ensemble. The marginal configuration space distribution is then $\rho(\boldsymbol{x}) \propto \exp[-S(\boldsymbol{x})]$, i.e. the distribution that we wanted to sample from.

An interesting question is what is the complete framework of possible ODE/SDE whose equilibrium solution corresponds to the target density $\rho(\boldsymbol{x}) \propto \exp[-S(\boldsymbol{x})]$. It has been argued (Ma et al., 2015) that the complete framework is given by a general form of the drift term $B(\boldsymbol{z}) = [D(\boldsymbol{z}) + Q(\boldsymbol{z})]\nabla H(\boldsymbol{z}) + \Gamma(\boldsymbol{z})$, where $H(\boldsymbol{z})$ is the Hamiltonian, $D(\boldsymbol{z})$ is positive definite diffusion matrix and $Q(\boldsymbol{z})$ is skew-symmetric matrix. $\Gamma(\boldsymbol{z})$ is specified by derivatives of $D(\boldsymbol{z})$ and $Q(\boldsymbol{z})$. This framework implicitly assumes that the equilibrium distribution is canonical on the phase space, $\rho(\boldsymbol{z}) \propto \exp[-H(\boldsymbol{z})]$. In general, however, we only need to require the marginal $\boldsymbol{x}$ distribution to be canonical, $\rho(\boldsymbol{x}) \propto \exp[-S(\boldsymbol{x})]$, giving rise to the possibility of additional formulations for which the stationary distribution matches the target distribution, but without the phase space distribution being canonical. One such general class of models is the Microcanonical Hamiltonian Monte Carlo (Robnik et al., 2022) (MCHMC), where the energy is conserved throughout the process, and a suitable choice of the Hamiltonian enforces the correct marginal configuration space distribution. MCHMC extends the deterministic dynamics (Ver Steeg and Galstyan, 2021) by adding momentum resampling, which is necessary for ergodicity. In this work, we will study the variable-mass MCHMC dynamics, which after rescaling time (coinciding with the inverse Sundman transformation (Skeel, 2009; Leimkuhler and Reich, 2004)) is the same dynamics as the isokinetic sampler (Evans et al., 1983; Tuckerman et al., 2001). In this paper, we explore the continuous-time limit, with and without diffusion. In Section 2 we derive the Liouville equation for continuous deterministic dynamics directly from the ODE, and show that its stationary solution is the target distribution. However, the deterministic dynamics of Tuckerman et al. (2001); Ver Steeg and Galstyan (2021) is not generically ergodic, even if additional variables are introduced as in Minary et al. (2003a,b), and even if it were, the convergence to equilibrium can be slow (Robnik et al., 2022). A possible solution is stochastic isokinetic dynamics (Leimkuhler et al., 2013), where the original phase space variables are coupled with $2d$ additional variables whose dynamics is stochastic, such that ergodicity can be rigorously established. However, the deterministic limit of this model no longer equals the deterministic isokinetic sampler and the variable mass MCHMC. In fact, it lives on a 3d-dimensional manifold, not 2d-1 dimensional manifold. With this approach, three additional hyperparameters are introduced, two masses and a damping parameter.

In section 3, we instead propose Microcanonical Langevin Monte Carlo (MCLMC), which directly adds stochasticity to the acceleration of the deterministic dynamics. This is a continuous-time analog of partially stochastically reorienting the velocity direction after every step of the deterministic dynamics, and is easy to integrate. MCLMC introduces one additional free parameter, the strength of the stochastic perturbation, whose optimal value can be determined by a short prerun (Robnik et al., 2022) and is anologus to the bounce rate in MCHMC. In contrast to the standard Langevin dynamics and stochastic isokinetic dynamics, the energy conservation in MCLMC leads to dynamics that does not

have a velocity damping term associated with the stochastic term, such that the noise is energy conserving. We derive the associated Fokker-Planck equation and show its stationary solution is the same as for the Liouville equation. In section 5 we prove that SDE is ergodic and in section 6 we demonstrate that it is also geometrically ergodic for smooth, log-convex target distributions. In section 7 we apply MCLMC to study the statistical $\phi^4$ field theory and compare it with Hamiltonian Monte Carlo.

## 2. Deterministic dynamics

We will study the ODE

$$\dot{\boldsymbol{x}} = \boldsymbol{u} \qquad \dot{\boldsymbol{u}} = P(\boldsymbol{u})\boldsymbol{f}(\boldsymbol{x}), \tag{1}$$

where $\boldsymbol{x}$ is the position of a particle in the configuration space and $\boldsymbol{u}$ is its velocity. $P(\boldsymbol{u}) \equiv I - \boldsymbol{u}\boldsymbol{u}^T$ is the projector to the direction perpendicular to the velocity and $\boldsymbol{f}(\boldsymbol{x}) \equiv -\nabla S(\boldsymbol{x})/(d-1)$ is the force[1]. One can arrive at this equation from at least two perspectives: (i) particle in external potential, being constrained to unit velocity, also known as the isokinetic ensemble (ii) energy-conserving dynamics of a particle with non-standard kinetic energy and external potential in natural time parameterization (Robnik et al., 2022; Ver Steeg and Galstyan, 2021), also known as the microcanonical ensemble.

The dynamics preserves the norm of $\boldsymbol{u}$ if we start with $\boldsymbol{u} \cdot \boldsymbol{u} = 1$,

$$\frac{d}{dt}(\boldsymbol{u} \cdot \boldsymbol{u}) = 2\boldsymbol{u} \cdot \dot{\boldsymbol{u}} = \boldsymbol{u} \cdot P(\boldsymbol{u})\boldsymbol{f} = (1 - \boldsymbol{u} \cdot \boldsymbol{u})(\boldsymbol{u} \cdot \boldsymbol{f}) = 0, \tag{2}$$

so the particle is confined to the $2d - 1$ dimensional manifold $\mathcal{M} = \mathbb{R}^d \times S^{d-1}$, i.e. the velocity is defined on a sphere of unit radius. We will denote the points on $\mathcal{M}$ by $\boldsymbol{z}$.

In this section we will briefly introduce the differential geometry formalism and show that $\exp\{-S(\boldsymbol{x})\}$ is the stationary distribution of Equation (1). In language of differential geometry, the dynamics of Equation (1) induces a flow on the manifold, which is a 1-parametrical family of maps from the manifold onto itself $\varphi_t : \mathcal{M} \to \mathcal{M}$, such that $\varphi_t(\boldsymbol{z})$ is the solution of Equation (1) with the initial condition $\boldsymbol{z}$. The flow induces the drift vector field $B(\boldsymbol{z})$, which maps scalar observables on the manifold $\mathcal{O}(\boldsymbol{z})$ to their time derivatives under the flow,

$$B(\boldsymbol{z})(\mathcal{O}) = \frac{d}{dt}\mathcal{O}\big(\varphi_t(\boldsymbol{z})\big)|_{t=0}. \tag{3}$$

We will be interested in the evolution of the probability density distribution of the particle under the flow. In differential geometry, the density is described by a volume form, which is a differential $(2d - 1)$-form,

$$\widehat{\rho}(\boldsymbol{z}) = \rho(\boldsymbol{z})\, dz^1 \wedge dz^2 \wedge ... dz^{2d-1}. \tag{4}$$

Volume form $\widehat{\rho}_t$ at time $t$ can be formally translated in time by the push-forward map $\varphi_{s*}$, $\widehat{\rho}_{t+s} = \varphi_{s*}\widehat{\rho}_t$. The infinitesimal form of the above equation gives us the differential equation

---

[1]. The force in Robnik et al. (2022); Ver Steeg and Galstyan (2021) was defined with a factor of $d$ rather than $d-1$, which required weights, proportional to $e^{-S(\boldsymbol{x})/d}$, meaning that the sampler without the weights converged to $e^{-S}/e^{-S/d}$. If we replace $S \to Sd/(d-1)$ this distribution becomes $e^{-S}$ and the weights are not required. This is equivalent to redefining the force as done here. In $d = 1$ this reweighting does not work and the original Hamiltonian formulation should be used.

for the density:

$$\frac{d}{dt}\widehat{\rho}_t = \frac{d}{ds}\big(\varphi_{s*}\widehat{\rho}_t\big)|_{s=0} = \frac{d}{ds}\big(\varphi^*_{-s}\widehat{\rho}_t\big)|_{s=0} \equiv -\mathcal{L}_B\widehat{\rho}_t = -\big(\mathrm{div}_{\widehat{\rho}_t}B\big)\widehat{\rho}_t, \tag{5}$$

which is also known as the Liouville equation. Here, $\varphi^*_{-s} = \varphi_{s*}$ is the pull-back map, $\mathcal{L}_B$ is the Lie derivative along the drift vector field $B$ and div is the divergence. This is the continuity equation for the probability in the language of differential geometry. The Liouville equation in coordinates is

$$\dot{\rho}(\boldsymbol{z}) = -\nabla \cdot \big(\rho B\big) \equiv -\sum_{i=1}^{2d-1}\frac{\partial}{\partial z^i}\big(\rho(\boldsymbol{z})B^i(\boldsymbol{z})\big) \tag{6}$$

We will work in the Euclidean coordinates $\{x^i\}_{i=1}^d$ on the configuration space and spherical coordinates $\{\vartheta^\mu\}_{\mu=1}^{d-1}$ for the velocities on the sphere, such that the manifold is parametrized by $\boldsymbol{z} = (\boldsymbol{x}, \boldsymbol{\vartheta})$. We will adopt the Einstein summation convention and use the Latin letters $(i, j, ...)$ to indicate the sum over the Euclidean coordinates and the Greek letters $(\mu, \nu, ...)$ for the sum over the spherical coordinates. The spherical coordinates are defined by the inverse transformation,

$$(u_1, u_2, \ldots u_d) = (\cos\vartheta^1,\ \sin\vartheta^1\cos\vartheta^2,\ \ldots,\ \sin\vartheta^1\cdots\sin\vartheta^{d-2}\cos\vartheta^{d-1},\ \sin\vartheta^1\cdots\sin\vartheta^{d-2}\sin\vartheta^{d-1}) \tag{7}$$

which automatically ensures $\boldsymbol{u}\cdot\boldsymbol{u} = 1$. The metric on the sphere in the spherical coordinates is

$$g_{\mu\nu} = \frac{\partial u_k}{\partial\vartheta^\mu}\frac{\partial u_k}{\partial\vartheta^\nu} = \mathrm{Diag}[1,\ \sin^2(\vartheta^1),\ \sin^2(\vartheta^1)\sin^2(\vartheta^2),\ ...]_{ij}, \tag{8}$$

and the volume element is $\sqrt{g} = \det g_{\mu\nu}^{1/2} = \Pi_{k=1}^{d-2}(\sin\vartheta^k)^{d-1-k}$. The drift vector field is

$$B = u_i(\boldsymbol{\vartheta})\frac{\partial}{\partial x^i} + \partial^\mu(\boldsymbol{u}\cdot\boldsymbol{f}(\boldsymbol{x}))\frac{\partial}{\partial\vartheta^\mu}, \tag{9}$$

where the second term results from $B_\nu = \frac{\partial u_i}{\partial\vartheta^\nu}P_{ij}f_j = \frac{\partial u_i}{\partial\vartheta^\nu}f_i$.

**Theorem 1** *The stationary solution of Liouville equation* (6) *is*

$$\rho_\infty \propto e^{-S(\boldsymbol{x})}\sqrt{g(\boldsymbol{\vartheta})}. \tag{10}$$

**Proof** Inserting $\rho_\infty$ in the Liouville equation (6) gives

$$\dot{\rho}_\infty = -\frac{\partial}{\partial x_i}(\rho_\infty B^i) - \frac{\partial}{\partial\vartheta^\mu}(\rho_\infty B^\mu) = -\boldsymbol{u}\cdot\partial_{\boldsymbol{x}}\rho_\infty - \frac{1}{\sqrt{g}}\partial_\mu\big(\sqrt{g}B^\mu\big)\rho_\infty.$$

The first term is $\boldsymbol{u}\cdot\partial_{\boldsymbol{x}}\rho_\infty = (d-1)\boldsymbol{u}\cdot\boldsymbol{f}\rho_\infty$. In the second term we recognize Laplacian of $\boldsymbol{u}\cdot\boldsymbol{f}$, so it transforms as a scalar under the transformations of the spherical coordinates. We can use this to simplify the calculation: at each fixed $\boldsymbol{x}$ we will pick differently oriented spherical coordinates, such that $\vartheta^1 = 0$ always corresponds to the direction of $\boldsymbol{f}(\boldsymbol{x})$ and $f_i = \delta_{1i}|\boldsymbol{f}|$. We then compute $B_\mu = (\partial_\mu u_1)|\boldsymbol{f}| = -\sin\vartheta^1\delta_{1\mu}|\boldsymbol{f}|$, so

$$\frac{1}{\sqrt{g}}\partial_\mu\big(\sqrt{g}B^\mu\big) = \frac{-|\boldsymbol{f}|}{\sin^{d-2}\vartheta^1}\frac{\partial}{\partial\vartheta^1}\sin^{d-1}\vartheta^1 = -(d-1)|\boldsymbol{f}|\cos\vartheta^1 = -(d-1)\boldsymbol{u}\cdot\boldsymbol{f}.$$

The last expression transforms as a scalar with respect to transformations on the sphere and is therefore valid in all coordinate systems, in particular, in the original one. Combining the two terms gives $\dot{\rho}_\infty = 0$, completing the proof. $\blacksquare$

## 3. Stochastic dynamics

In Robnik et al. (2022) it was proposed that adding a random perturbation to the momentum direction after each step of the discretized deterministic dynamics boosts ergodicity, but the continuous-time version was not explored. Here, we consider a continuous-time analog and show this leads to Microcanonical Langevin SDE for the particle evolution and to Fokker-Planck equation for the probability density evolution. We promote the deterministic ODE of Equation (1) to the following Microcanonical Langevin SDE:

$$d\boldsymbol{x} = \boldsymbol{u}dt \tag{11}$$
$$d\boldsymbol{u} = P(\boldsymbol{u})\boldsymbol{f}(\boldsymbol{x})dt + \eta P(\boldsymbol{u})d\boldsymbol{W}.$$

Here, $\boldsymbol{W}$ is the Wiener process, i.e. a vector of random noise variables drawn from a Gaussian distribution with zero mean and unit variance, and $\eta$ is a free parameter. The last term is the standard Brownian motion increment on the sphere, constructed by an orthogonal projection from the Euclidean $\mathbb{R}^d$, as in Elworthy (1998a).

More formally, we may write Equation (11) as a Stratonovich degenerate diffusion on the manifold (Elworthy, 1998b; Baxendale, 1991; Kliemann, 1987),

$$d\boldsymbol{z} = B(\boldsymbol{z})dt + \sum_{i=1}^{d} \sigma_i(\boldsymbol{z}) \circ dW_i, \tag{12}$$

where $W_i$ are independent $\mathbb{R}$-valued Wiener processes and $\sigma_i$ are vector fields, in coordinates expressed as

$$\sigma_i(\boldsymbol{\vartheta}) = g^{\mu\nu}(\boldsymbol{\vartheta})\frac{\partial u_i}{\partial \vartheta^\nu}(\boldsymbol{\vartheta})\frac{\partial}{\partial \vartheta^\mu}(\boldsymbol{\vartheta}). \tag{13}$$

With the addition of the diffusion term, the Liouville equation (6) is now promoted to the Fokker-Planck equation (Elworthy, 1998a),

$$\dot{\rho} = -\nabla \cdot (\rho B) + \frac{\eta^2}{2}\widehat{\nabla}^2\rho, \tag{14}$$

where $\widehat{\nabla}^2 = \nabla^\mu\nabla_\mu$ is the Laplace-Beltrami operator on the sphere and $\nabla_\mu$ is the covariant derivative on the sphere. In coordinates, the Laplacian can be computed as $\frac{1}{\sqrt{g}}\partial_\mu(\sqrt{g}\partial^\mu\rho)$.

**Theorem 2** *The distribution $\rho_\infty$ of Equation (10) is a stationary solution of the Fokker-Planck equation (14) for any value of $\eta$.*

**Proof** Upon inserting $\rho_\infty$ into the right-hand-side of the Fokker-Planck equation, the first term vanishes by Theorem 1. The second term also vanishes,

$$\widehat{\nabla}^2\rho_\infty \propto \nabla^\mu\nabla_\mu\sqrt{g}e^{-S(\boldsymbol{x})} = e^{-S(\boldsymbol{x})}\nabla^\mu\nabla_\mu\sqrt{g} = 0,$$

since the covariant derivative of the metric determinant is zero. $\blacksquare$

## 4. Discretization

Consider for a moment Equation (11) with only the diffusion term on the right-hand-side. This SDE describes the Brownian motion on the sphere and the identity flow on the $\boldsymbol{x}$-space (Elworthy, 1998a). Realizations from the Brownian motion on the sphere can be generated exactly (Li and Erdogdu, 2020). Let us denote by $\psi_s^\eta$ the corresponding density flow map, such that $\psi_s^\eta[\rho_t] = \rho_{t+s}$. The flow of the full SDE (11) can then be approximated at discrete times $\{n\epsilon\}_{n=0}^\infty$ by the Euler-Maruyama scheme (Øksendal and Øksendal, 2003):

$$\rho_{(n+1)\epsilon} = \psi_\epsilon^\eta[\varphi_{\epsilon*}\, \rho_{n\epsilon}]. \tag{15}$$

For a generic SDE, this approximation leads to bias in the stationary distribution. This is however not the case in MCLMC:

**Theorem 3** *The distribution $\rho_\infty$ of Equation (10) is preserved by the Euler-Maruyama scheme (15) for any value of $\eta$.*

**Proof** The deterministic push forward map preserves $\rho_\infty$ by the Theorem 1. The Fokker-Planck equation for the stochastic-only term is $\dot{\rho} = \frac{\eta^2}{2}\widehat{\nabla}\rho$ which preserves $\rho_\infty$ by the Theorem 2. ∎

In the standard Langevin equation, the fluctuation term is accompanied by a dissipation term, and the strength of both is controlled by damping coefficient. The deterministic and stochastic parts do not preserve the stationary distribution separately. In contrast, for MCLMC an exact deterministic ODE integrator would remain exact with SDE, so the integration scheme for the deterministic dynamics is the only bias source.

Furthermore, in the discrete scheme (15) it is not necessary to have the Brownian motion on the sphere as a stochastic update in order to have $\rho_\infty$ as a stationary distribution. In fact, any discrete stochastic process on the sphere, which has the uniform distribution as the stationary distribution will do, for example the one used in Robnik et al. (2022).

## 5. Ergodicity

We have established that $\rho_\infty$ is the stationary distribution of the MCLMC SDE. Here we demonstrate the uniqueness of the stationary distribution.

Let's define $\mathcal{S}(\boldsymbol{z}) = \{B(\boldsymbol{z}), \sigma_1(\boldsymbol{z}), \sigma_2(\boldsymbol{z}), \ldots \sigma_d(\boldsymbol{z})\}$. The Hörmander's condition (Leimkuhler and Matthews, 2015) is satisfied at $\boldsymbol{z} \in \mathcal{M}$ if the smallest Lie algebra containing $\mathcal{S}(\boldsymbol{z})$ and closed under $v \mapsto [B, v]$ is the entire tangent space $T_{\boldsymbol{z}}(\mathcal{M})$. Here $[\cdot, \cdot]$ is the Lie bracket, in coordinates $[X, Y] = X^i \partial_i Y^k \partial_k - Y^i \partial_i X^k \partial_k$.

**Lemma 4** *(Hörmander's property) MCLMC SDE satisfies the Hörmander's condition for all $\boldsymbol{z} \in \mathcal{M}$.*

**Proof** Fix some $\boldsymbol{z} = (\boldsymbol{x}, \boldsymbol{u})$. The tangent space at $\boldsymbol{z}$ is a direct sum $T_{\boldsymbol{z}}(\mathcal{M}) = T_{\boldsymbol{x}}(\mathbb{R}^d) \oplus T_{\boldsymbol{u}}(S^{d-1})$. $\sigma_i$ span $T_{\boldsymbol{u}}(S^{d-1})$ by construction: they were obtained by passing the basis of $\mathbb{R}^d$ through the orthogonal projection, $P(\boldsymbol{u})$, which has rank $d-1$. For convenience we may further decompose $T_{\boldsymbol{x}}(\mathbb{R}^d) = U \oplus U^\perp$, where $U = \{\lambda \boldsymbol{u} | \lambda \in \mathbb{R}\}$ is the space spanned by $\boldsymbol{u}$

and $U^\perp$ is its orthogonal complement. We can write $B = B_x + B_u$, such that $B_x \in T_{\boldsymbol{x}}(\mathbb{R}^d)$ and $B_u \in T_{\boldsymbol{u}}(S^{d-1})$, see also Equation (9). $B_x = u_i \partial_{x^i}$ spans $U$, so we are left with covering $U^\perp$. $[B_u, \sigma_i] \in T_{\boldsymbol{u}}(S^{d-1})$, so it does not interest us anymore. However,

$$[B_x,\, \sigma_i] = -\sigma_i(B_x) = g^{\mu\nu} \frac{\partial u_i}{\partial \vartheta^\mu} \frac{\partial u_j}{\partial \vartheta^\nu} \partial_j,$$

so $[B_x,\, \sigma_i]$ span $U^\perp$, completing the proof. ∎

**Lemma 5** *(Path accessibility of points): For every two points $\boldsymbol{z}_i, \boldsymbol{z}_f \in \mathcal{M}$ there exists a continuous path $\gamma : [0, T] \to \mathcal{M}$, $\gamma(0) = \boldsymbol{z}_i$, $\gamma(T) = \boldsymbol{z}_f$ and values $0 = t_0 < t_1 < \cdots < t_N = T$ with a corresponding sequence of vectors $v_n \in \mathcal{S}$, such that for each $0 \leq n < N$, $\dot{\gamma}(t) = v_n(\gamma(t))$ for $t_n < t < t_{n+1}$.*

**Proof** Starting at $\boldsymbol{z}_i$, we will reach $\boldsymbol{z}_f$ in three stages. In stage I, we will use $\sigma_i$ to reorient $\boldsymbol{u}_i$ to the desired direction $\bar{\boldsymbol{u}}$ (as determined by the stage II). In stage II we will then follow $B$ to reach the final destination $\boldsymbol{x}_f$ in the configuration space. In stage III, we will reorient the velocity from the end of stage II to the desired $\boldsymbol{u}_f$.

  Stage I: First we note that $\sigma_k = -\sin\vartheta^k \frac{\partial}{\partial\vartheta^k}$ if $\vartheta^i = \pi/2$ for all $i < k$. In this case, $\dot{\gamma} = \sigma_k$ has a solution $t = t_0 + \log\frac{\tan\vartheta^l(t)/2}{\tan\vartheta^l(t_0)/2}$ and keeps $\vartheta^l(t) = \vartheta^l(t_0)$ for $l \neq k$. This means that we can first recursively set $\vartheta^n$ to $\pi/2$ by selecting $t_{n+1} = t_n + \log\frac{\tan\pi/4}{\tan\vartheta^n_i/2}$ and $v_n = \sigma_n$ for $n = 1, 2, \ldots d-1$. Then we go back and set all $\theta^n$ to their desired final value, by selecting $t_{n+1} = t_n + \frac{\tan\bar{\vartheta}^{2d-1-n}/2}{\tan\pi/4}$ for $n = d, d+1, \ldots 2(d-1)$.

  Stage II: As shown in Robnik et al. (2022), up to time rescaling, the trajectories of (1) (flows under $B$) are the trajectories of the Hamiltonain $H = |\Pi^2|/m(S(\boldsymbol{x}))$ and are in turn also the geodesics on a conformally flat manifold (Robnik et al., 2022). Any two points can be connected by a geodesic and therefore $B$ connects any two points $\boldsymbol{x}_i$ and $\boldsymbol{x}_f$.

  Stage III: use the program from stage I. ∎

**Lemma 6** *(Smooth, nonzero density) The law of $\boldsymbol{z}(t)$ admits a smooth density. For every $\boldsymbol{z}_i \in \mathcal{M}$ and every Lebesgue-positive measure Borel set $\mathcal{U}_f \in B(\mathcal{M})$, there exists $T(\boldsymbol{z}_i, \mathcal{U}_f) \geq 0$, such that $P(\boldsymbol{z}(T) \in \mathcal{U}_f | \boldsymbol{z}(0) = \boldsymbol{z}_i) > 0$.*

**Proof** Fix $\boldsymbol{z}_f \in \mathcal{M}$, such that every neighborhood of $\boldsymbol{z}_f$ has a positive-measure intersection with $\mathcal{U}_f$. By the Hörmander's theorem, Lemma 4 implies that there exist $t_1 > 0$, $m > 0$ and a non-empty open subset $\mathcal{U}_i$ of a chart on $\mathcal{M}$, such that the law of $\boldsymbol{z}(t_1)$ has a Lebesgue density of at least $m$ on $\mathcal{U}_i$. Now let $t_2$ be the time $T$ from Lemma 5, when applied to $\boldsymbol{z}_f$ and any point in $\mathcal{U}_i$. $T$ from this theorem will be $t_1 + t_2$. For k = 1, 2 let $(I_k, \mathcal{I}_k, \mu_k)$ be the Wiener space defined over $[0, t_k]$ and $\Phi_k$ be the time-$t_k$ mappings of the SDE. Since $\mathcal{U}_i$ is open, applying the "support theorem" (Theorem 3.3.1(b) of Baras et al. (1990)) to the reverse-time SDE gives that $\mu_2(\boldsymbol{z}_f \in \Phi_2(\mathcal{U}_i)) > 0$. Hence the Lemma 10 gives the desired result. ∎

**Theorem 7** *(ergodicity) MCLMC SDE* (11) *admits a unique stationary distribution.*

**Proof** This follows immediately from Lemma 6 and Theorem 6 in Noorizadeh (2010). ∎

## 6. Geometric ergodicity

Geometric ergodicity is a statement that the convergence to the stationary distribution is exponentially fast. In this section we will assume that the target $S(\boldsymbol{x})$ is $M$-smooth and $m$-convex meaning that $mI < \partial_{ij}S(\boldsymbol{x}) < MI$. As in Leimkuhler and Matthews (2015) we will also assume periodic boundary conditions at large $\boldsymbol{x}$, implying that the gradient is bounded, $\nabla S(\boldsymbol{x}) < g_{\max}$.

Let $\mathcal{L}$ be the infinitesimal generator of the MCLMC SDE:

$$\mathcal{L}\phi = B(\phi) + \frac{\eta^2}{2}\widehat{\nabla}^2\phi. \tag{16}$$

**Lemma 8** *(Lyapunov function)* $\phi(\boldsymbol{x}, \boldsymbol{u}) = \boldsymbol{u} \cdot \nabla S(\boldsymbol{x}) + g_{\max}$ *is a Lyapunov function:*

- $\phi(\boldsymbol{z}) > 0$

- $\phi(\boldsymbol{z}) \to \infty$ *as* $|\boldsymbol{z}| \to \infty$.

- $\mathcal{L}\phi < -a\phi + b$ *for some* $a, b > 0$.

**Proof** The first property follows by

$$\boldsymbol{u} \cdot \nabla S > -|\boldsymbol{u}||\nabla S| > -g_{\max}.$$

The second property is trivially satisfied because the phase space is bounded.

For the third property, let's compute terms one by one:

- The $\boldsymbol{x}$-part of the drift gives $\boldsymbol{u} \cdot \partial_{\boldsymbol{x}}\phi = u_i u_j \partial_{ij}S(\boldsymbol{x}) < M|\boldsymbol{u}|^2 = M$.

- The $\boldsymbol{u}$-part of the drift gives $\partial^\mu(\boldsymbol{u} \cdot \boldsymbol{f}(\boldsymbol{x}))\partial_\mu(\boldsymbol{u} \cdot \nabla S(\boldsymbol{x})) = -\frac{1}{d-1}|\partial_\nu(\boldsymbol{u} \cdot \nabla S)|_g^2 \leq 0$ where $|v|_g^2 = g^{\mu\nu}v_\mu v_\nu$ is the metric induced norm.

- The Laplacian of $u_1 = \cos\vartheta$ is

$$\widehat{\nabla}^2 u_1 = \frac{1}{\sqrt{g}}\partial_\mu(\sqrt{g}\partial^\mu u_1) = \frac{1}{(\sin\theta)^{\frac{d}{2}-1}}\left((\sin\theta)^{\frac{d}{2}-1}(\cos\theta)'\right)' = -\frac{d}{2}\cos\vartheta = -\frac{d}{2}u_1$$

so by symmetry $\widehat{\nabla}^2\boldsymbol{u} = -\frac{d}{2}\boldsymbol{u}$ and $\widehat{\nabla}^2\phi = -\frac{d}{2}\boldsymbol{u} \cdot \nabla S$.

Combining everything together, we get:

$$\mathcal{L}\phi < M - \frac{\eta^2 d}{4}\boldsymbol{u} \cdot \nabla S = -a\phi + b$$

for $a = \eta^2 d/4$ and $b = M + ag_{\max}$. ∎

With Lyapunov function in hand the result is immediate:

**Theorem 9** *(Geometric ergodicity) There exist constants $C > 0$ and $\lambda > 0$, such that for all observables $\mathcal{O}(z)$ for which $|\mathcal{O}(z)| < \phi(z)$ the expected values $\langle\mathcal{O}\rangle_\rho = \int \mathcal{O}(z)\rho(z)dz$ under the MCLMC SDE* (11) *converge at least exponentially fast:*

$$|\langle\mathcal{O}\rangle_{\rho(t)} - \langle\mathcal{O}\rangle_{\rho_\infty}| < Ce^{-\lambda t}\phi(z_0). \tag{17}$$

**Proof** This follows directly from theorem 6.2 in Leimkuhler and Matthews (2015), given the Lyapunov function from Lemma 8 and smooth positive density from Lemma 6. ∎

## 7. Applications

To show the promise of MCLMC as a general purpose MCMC tool, we apply it to the scalar $\phi^4$ field theory in two Euclidean dimensions and to benchmark hierarchical Bayesian inference problems.

In the discrete scheme (15) it is not necessary to have the Brownian motion on the sphere as a stochastic update in order to have $\rho_\infty$ as a stationary distribution. In fact, any discrete stochastic process on the sphere, which has the uniform distribution as the stationary distribution and acts as an identity on the $x$-space will do. In the practical algorithm, we therefore avoid generating the complicated Brownian motion on the sphere and instead use the generative process $u_{(n+1)\epsilon} = (u_{n\epsilon} + \nu r)/|u_{n\epsilon} + \nu r|$, where $r$ is a random draw from the standard normal distribution and $\nu$ is a parameter with the same role as $\eta$. We tune the parameter $\eta$ by estimating the effective sample size (ESS) (Robnik et al., 2022). We approximate the deterministic flow $\varphi_t$ with the Minimal Norm integrator (Omelyan et al., 2003; Robnik et al., 2022) and tune the step size by targeting a predefined energy error variance per dimension, as in (Robnik et al., 2022). For $\phi^4$ field theory, the tuning of the step size and $\eta$ is done at each $\lambda$ level separately and is included in the sampling cost. It amounts to around 10% of the sampling time.

### 7.1. Lattice $\phi^4$ field theory

This is one of the simplest non-trivial lattice field theory examples. The scalar field in a continuum is a scalar function $\phi(x, y)$ on the plane with the area $V$. The probability density on the field configuration space is proportional to $e^{-S[\phi]}$, where the action is

$$S[\phi(x, y)] = \int \left( -\phi\, \partial^2\phi + m^2\phi^2 + \lambda\phi^4 \right)dxdy. \tag{18}$$

The squared mass $m^2 < 0$ and the quartic coupling $\lambda > 0$ are the parameters of the theory. The system is interesting as it exhibits spontaneous symmetry breaking, and belongs to the same universality class as the Ising model. The action is symmetric to the global field flip symmetry $\phi \to -\phi$. However, at small $\lambda$, the typical set of field configurations splits in two symmetric components, each with non-zero order parameter $\langle\bar{\phi}\rangle$, where $\bar{\phi} = \frac{1}{V}\int\phi(x,y)dxdy$ is the spatially averaged field. The mixing between the two components is highly unlikely, and so even a small perturbation can cause the system to acquire non-zero order parameter. One such perturbation is a small external field $h$, which amounts to the additional term

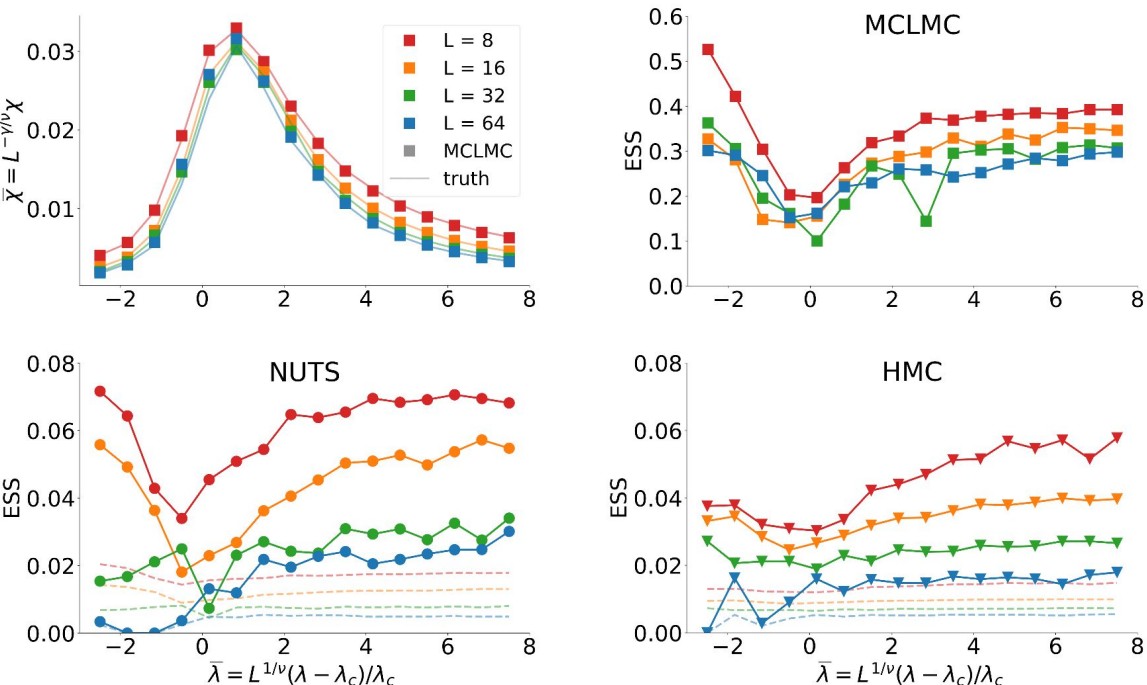

Figure 1: Top left: susceptibility in the vicinity of the phase transition. We follow Gerdes et al. (2022) and rescale the susceptibility and the quartic coupling by the Ising model critical exponents $\nu = 1$, $\gamma = 7/4$ and the critical coupling $\lambda_C = 4.25$ from Vierhaus (2010). The rescaling removes most of the lattice size dependence (Goldenfeld, 2018). MCLMC agrees with the truth (obtained by a very long NUTS run). Top right: effective sample size (ESS) per action gradient evaluation for MCLMC. Higher is better. MCLMC tuning cost is included. Bottom: same for NUTS and HMC. The dotted lines are the corresponding results if the tuning cost of 500 warm-up samples is taken into account.

$-h \int \phi(x, y) dx dy$ in the action. The susceptibility of the order parameter to an external field is defined as

$$\chi = V \frac{\partial \bar{\phi}}{\partial h}|_{h=0} = \lim_{h \to 0^+} V \langle (\bar{\phi} - \langle \bar{\phi} \rangle)^2 \rangle, \tag{19}$$

which diverges at the critical point, where the second order phase transition occurs.

The $\phi^4$ theory does not admit analytic solutions due to the quartic interaction term. A standard approach is to discretize the field on a lattice and make the lattice spacing as fine as possible (Gattringer and Lang, 2009). The field is then specified by a vector of field values on a lattice $\phi_{ij}$ for $i, j = 1, 2, \ldots L$. The dimensionality of the configuration space is $d = L^2$. We will impose periodic boundary conditions, such that $\phi_{i,L+1} = \phi_{i1}$ and

$\phi_{L+1,j} = \phi_{1j}$. The $h = 0$ lattice action is (Vierhaus, 2010)

$$S_{\text{lat}}[\phi] = \sum_{i,j=1}^{L} 2\phi_{ij}\big(2\phi_{ij} - \phi_{i+1,j} - \phi_{i,j+1}\big) + m^2 \phi_{ij}^2 + \lambda \phi_{ij}^4. \tag{20}$$

As common in the literature (Albergo et al., 2019, 2021; Gerdes et al., 2022), we will fix $m^2 = -4$ (which removes the diagonal terms $\phi_{ij}^2$ in the action) and study the susceptibility as a function of $\lambda$. The susceptibility estimator is $\chi = L^2 \langle \big(\bar{\phi} - \langle\bar{\phi}\rangle\big)^2 \rangle$, where $\bar{\phi} = \frac{1}{L^2} \sum_{ij} \phi_{ij}$, and the expectation $\langle \cdot \rangle$ is over the samples (Gerdes et al., 2022).

A measure of the efficiency of sampling performance is the number of action gradient calls needed to have an independent sample. Often we wish to achieve some accuracy of expected second moments (Robnik et al., 2022). We define the squared bias $b_2^2$ as the relative error of the expected second moments in the Fourier basis, $b_2^2 = \frac{1}{L^2} \sum_{k,l=1}^{L} \big(1 - \frac{\langle |\widetilde{\phi}_{kl}|^2 \rangle_{\text{sampler}}}{\langle |\widetilde{\phi}_{kl}|^2 \rangle_{\text{truth}}}\big)^2$, where $\widetilde{\phi}$ is the scalar field in the Fourier basis, $\widetilde{\phi}_{kl} = \frac{1}{\sqrt{L^2}} \sum_{nm=1}^{L} \phi_{nm} e^{-2\pi i(kn+lm)/L}$. In analogy with Gaussian statistics, we define the effective sample size to be $2/b_2^2$. Here, we report the effective sample size per action gradient evaluation at the instant when $b_2 = 0.1$, which corresponds to 200 effective samples. The number we report is ESS per action gradient evaluation, such that its inverse gives the number of gradients needed to achieve one independent sample.

In Figure 1 we compare MCLMC to standard HMC (Duane et al., 1987) and to a self-tuned HMC variant NUTS (Hoffman et al., 2014), both implemented in NumPyro (Phan et al., 2019). For HMC, we find that the optimal number of gradient calls between momentum resamplings to be 20, 30, 40 and 50 for lattice sizes $L = $ 8, 16, 32 and 64. The step size is determined with the dual averaging algorithm, targeting acceptance rate of 0.8 (NumPyro default), which adds considerably to the overall cost (Figure 1).

For all samplers, we use an annealing scheme, starting at high $\lambda$ and using the final state of the sampler as an initial condition at the next lowest $\lambda$ level. The initial condition at the highest $\lambda$ level is a random draw from the standard normal distribution on each lattice site. There is a near perfect agreement between a very long NUTS run (denoted as truth) and MCLMC in terms of susceptibility. Above the phase transition ($\bar{\lambda} \gtrsim 1$), ESS for MCLMC and HMC is relatively constant with $\bar{\lambda}$. ESS for NUTS and HMC scales with $L$ as $d^{-1/4} = L^{-1/2}$, as expected from adjusted HMC (Neal et al., 2011). At the phase transition, NUTS and HMC suffer from the critical slowing down, resulting in lower ESS. In contrast, ESS for MCLMC is almost independent of $\bar{\lambda}$ and $L$. Overall, MCLMC outperforms HMC and NUTS by 10-100 at $L = 64$ if HMC and NUTS tuning is not included, and by at least 40 if tuning is included (MCLMC auto-tuning is cheap and included in the cost, and we use the recommended 500 warm up samples for tuning of NUTS and HMC). We thus expect that for $d = 10^8$, typical of state-of-the-art lattice quantum chromodynamics calculations, the advantage of MCLMC over HMC and NUTS will be 2–3 orders of magnitude due to $d^{1/4}$ scaling. MCLMC also significantly outperforms recently proposed Normalizing Flow (NF) based samplers (Albergo et al., 2019; Gerdes et al., 2022). NFs scale poorly with dimensionality, and the training time increases by about one order of magnitude for each doubling of $L$, e.g. of order 10 hours for $L = 32$ to reach 90% acceptance, and 60 hours to reach 60% acceptance at $L = 64$ (Gerdes et al., 2022). In contrast, the wall-clock time of

MCLMC at $L = 64$ on a GPU is a fraction of a second, while even at $L = 8096$ (completely out of reach of current NF based samplers) it is only 15 seconds.

## 7.2. Hierarchical Bayesian inference

We here test MCLMC on two Bayesian inference problems, taken from the Inference gym (Sountsov et al., 2020). Brownian Motion is a 32 dimensional problem, where Brownian motion with unknown innovation noise is fitted to the noisy and partially missing data. Item Response theory is a 501 dimensional hierarchical problem where students' ability is inferred, given their test results. We follow Hoffman and Sountsov (2022) and define the error of the expectation value of $f(\boldsymbol{x})$ as $b_f^2 = (\langle f \rangle_{\text{sampler}} - \langle f \rangle)^2 / \text{Var}[f]$, where ground truth expectation values $\langle f \rangle$ and $\text{Var}[f] = \langle (f - \langle f \rangle)^2 \rangle$ are computed by very long NUTS runs. We will measure samplers' efficiency as the number of gradient evaluations needed to achieve low average second moment error $b^2 \equiv \sum_{i=1}^{d} b_{x_i^2}^2 / d < 0.01$, where averaging was performed over parameters of the model and we also average over 128 independent chains. The results are shown in Table 1.

| problem | MCLMC | NUTS |
|---|---|---|
| Brownian Motion | **2032** | 6369 |
| Item response theory | **3312** | 11140 |

Table 1: Number of gradient evaluations to low bias, lower value is better. MCLMC outperforms NUTS by more than a factor of three in both examples.

## 8. Conclusions

We introduced an energy conserving stochastic Langevin process in the continuous time limit that has no damping, derived the corresponding Fokker-Planck equation. Its equilibrium solution is microcanonical in the total energy, yet its space distribution equals the desired target distribution given by the action, showing that the framework of Ma et al. (2015) is not a complete recipe of all SDEs whose equilibrium solution is the target density. We have also proven ergodicity demonstrating that the stationary solution is unique, and geometric ergodicity, demonstrating that the convergence to the stationary distribution is exponentially fast.

MCLMC is also of practical significance: we show it vastly outperforms the state-of-the-art HMC on a lattice $\phi^4$ model. In lattice quantum chromodynamics (Gattringer and Lang, 2009; Degrand and DeTar, 2006) the computational demands are particularly intensive, and numerical results presented here suggest that MCLMC could offer significant improvements over HMC in the setting of high dimensional models. We have also demonstrated that MCLMC offers significant (factor of three) improvements over the state-of-the-art algorithm NUTS (a variant of HMC) on hierarchical Bayesian inference problems, suggesting MCLMC outperforms HMC/NUTS over a wide range of problems.

## Acknowledgments

We thank Julian Newman for proving lemmas 6 and 10 and Qijia Jiang for useful discussions. This material is based upon work supported in part by the Heising-Simons Foundation grant 2021-3282 and by the U.S. Department of Energy, Office of Science, Office of Advanced Scientific Computing Research under Contract No. DE-AC02-05CH11231 at Lawrence Berkeley National Laboratory to enable research for Data-intensive Machine Learning and Analysis.

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

## Appendix A.

We here include a supplementary lemma, needed in the proof of lemma 6:

**Lemma 10** *Let $M$ be a $C^1$ manifold. Let $(I, \mathcal{I}, \mu)$ and $(J, \mathcal{J}, \nu)$ be probability spaces, and over these probability spaces respectively, let $(\Phi_\alpha)_{\alpha \in I}$ and $(\Psi_\beta)_{\beta \in J}$ be random $C^1$ self-embeddings of $M$. Fix $x_1, x_2 \in M$, let $p$ be the law over $\mu$ of $\alpha \mapsto \Phi_\alpha(x_1)$, and let $\tilde{p}$ be the law over $\mu \otimes \nu$ of $(\alpha, \beta) \mapsto \Psi_\beta(\Phi_\alpha(x_1))$. Suppose there exists $m > 0$ and an open subset $U$ of a chart on $M$ such that*

- *for every $A \in \mathcal{B}(U)$, $p(A) \geq m \operatorname{Leb}(A)$;*

- *$\nu(\beta \in J | x_2 \in \Psi_\beta(U)) > 0$.*

*Then there exists $\tilde{m} > 0$ and a neighborhood $\tilde{U}$ of $x_2$ contained in a chart on $M$ such that for every $A \in \mathcal{B}(\tilde{U})$, $\tilde{p}(A) \geq \tilde{m} \operatorname{Leb}(A)$.*

**Proof** One can find a $\nu$-positive measure set $J' \subset J$, a neighborhood $\tilde{U}$ of $x_2$ contained in a chart on $M$, and a value $r > 0$, such that for all $\beta \in J'$ and $x \in \tilde{U}$, we have $\Psi_\beta^{-1}(x) \in U$ and $|\det(D(\Psi_\beta^{-1})(x))| \geq r$. Now take any $A \in \mathcal{B}(\tilde{U})$ and let

$$E = \{(\alpha, \beta) \in I \times J' | \Psi_\beta(\Phi_\alpha(x_1)) \in A\}$$

then

$$\tilde{p}(A) \geq (\mu \otimes \nu)(E) = \int_{J'} \mu(\alpha \in I | \Psi_\beta(\Phi_\alpha(x_1)) \in A) \, \nu(d\beta) = \int_{J'} p(\Psi_\beta^{-1}(A)) \, \nu(d\beta)$$

$$\geq m \int_{J'} \text{Leb}(\Psi_\beta^{-1}(A)) \, \nu(d\beta) \geq m\nu(J')r = \tilde{m} \, \text{Leb}(A).$$

Therefore, $\tilde{p}(A) \geq \tilde{m} \, \text{Leb}(A)$. ∎

