# OpenReview forum: "Fluctuation without dissipation: Microcanonical Langevin Monte Carlo"
_approximateinference.org/AABI/2024/Symposium_Archival_Track — AABI 2024 - Archival Track_

### Official Review · Reviewer_VURU · 2024-04-19
**An exciting new direction for MCMC**

**Rating:** 8
**Confidence:** 2

**Review:**

### **Summary**

This paper considers an intriguing kind of MCMC move to help us sample from a potential S:
* from an initial value, $x$,
* select a random momentum, $u \sim \mathrm{Uniform}(\mathbb{S}^{d-1})$ (or possibly from the d-2 dimensional sphere of points orthogonal to the gradient of S, I couldn't tell),
* evolve the SDE $dx_t=u_t$ and $d u_t=-(I-u_tu_t^\top) \nabla S(x_t)dt/(d-1) + \eta (I-u_tu_t^\top) dW_t$,
* and return the new value of $x$.

They refer to this move as MCLMC.  However, there's a move called MCLMC in a JMLR paper (Robnik 2023) which seems similar (maybe identical?). I'll refer to the move studied in this paper as MCLMC2024.  It has some HMC qualities to it (but different because the momentum is always unit norm and there's a diffusion term).  It's perhaps closest to underdamped langevin (but different because momentum always has unit norm).

MCLMC2024 seems to be pretty good.  It performs well in a few experiments.  The paper also goes into some theoretical properties of the move.

### **Contributions**


Overall, this work seems to be somewhat redundant with Robnik 2023.  Yet it seems to me nonetheless valuable to include this work in this new venue: the new work has to potential to present some of the ideas in the old work in a more concise and accessible fashion.  Also I think there is at least some new content here, though I'm not 100% confident about that.  In brief, I think the main contributions here are...
* Digesting (Robnik 2023) into into a single choice of algorithm (MCLMC2024) that might be quite useful in practice and probably should be better known (I certainly hadn't heard about it).
* Developing some (new?) theory about the MCLMC2024.
* Running a few experiments about MCLMC2024.

Within the theory contributions, it was difficult for me to discern what was new and what was old.  Is Theorem 1 known?  Theorem 2 is of course something we want to see (but was it not already shown in Robnik 2023?).  Theorem 3 is presumably new; its fun that this Langevin-ish sort of algorithm actually doesn't have to worry about any discretization error besides those of the deterministic dynamics. Theorems 7 and 9 are good to have...I'm assuming they're new?  In light of Theorem 2, Theorems 7 and 9 are perhaps not very surprising (the authors assume that the magnitude of the drift is bounded and periodic boundary conditions, so nothing can really go off the rails), but it is good to have such theorems anyway.  They add a reassuring solidity to the whole framework.

### **Concerns**

* I did not understand whether a metropolis-hastings (MH) adjustment step was needed.  It seems like authors are approximately sampling from an SDE.  In the case of overdamped langevin, many practitioners follow their SDE steps with a MH accept/reject step (i.e., they perform MALA).  In the case of HMC, an MH accept/reject step is also generally considered best practice.  Authors might want to do that here?  Since the deterministic dynamics are not, in fact, calculated exactly?  So, are the authors doing that?  Or not?   If they aren't, it would be nice to discuss whether the adjustment is easy to compute if it is desired.  I think the discretized dynamics aren't generally symmetric (i.e., for most integrators, if you simulate forward, reverse momentum, and then simulate again, you don't get back where you started).  So possibly it is very difficult to compute the appropriate MH adjustment?  Or is it easy if you use the leapfrog integrator from the ESH paper?  I'm not actually clear if that ESH leapfrog integrator actually has the necessary properties to compute the MH adjustment (i.e., the forward step is invertible and the Jacobian is easy to compute).
* Relatedly, especially without an MH step, I'm worried about what happens when $\eta$ is small.  It seems like things could be very numerically unstable and really fail to model the target (I ran some simulations with the $\eta=0$ case and it appeared to me that at least with a simple leapfrog integrator I got results that didn't make sense to me).  I'm not sure that the automatic $\eta$ tuning will necessary select a sufficiently large $\eta$.  This seems like a concern worth investigating and alerting the reader to: in my understanding, without some additional care, It is possible that the method could fail (and fail badly!) to sample from the target distribution, even in the asymptotic limit, due to discretization artifacts.
* It's worth mentioning this doesn't work at all if d is one and u is constrained to lie in the disconnected set {-1,1}?  Indeed, I guess you get a divide by zero error in the formula.

### **Suggestions**

It is of course always hard to know who your audience is.  I found the flow confusing.

If the audience was me, here's what I'd suggest for an organization.

* In the introduction, explain the problem the authors are trying to solve (i.e., sampling from distributions).  Then explain why existing methods aren't so hot.  Then explain, broadly, the promise of the MCLMC2024 approach.
* Then explain formally what (MCLMC2024) is.  Clarify its historical relationship to moves that other papers have tried (if it has been tried before in Robnik 2023, then say that).  If the deterministic version ($\eta=0$) is guaranteed to stay on the same level set $\{x:\ S(x)=c\}$ over the evolution, then say that.  Otherwise, if there is no such energy, explain more clearly what it has to do with microcanonical hmc (in which, in my understanding, energy is usually conserved).
* Talk about the theory that the authors have in hand.
* Run some "easy" simulations (see below).
* Show experiments (more experiments if authors have more time!).  The plot was also pretty impossible to read, because the ESS for all the different methods were on different scales.  It was very hard to see which method was doing the best for any given value of $\bar \lambda$.  I think authors were trying to support this claim that HMC/NUTS performance was bad around the critical temp, but I couldn't how I was supposed to see that anyway.  Also I didn't understand why MCLMC didn't have dotted lines (doesn't it have to take time to tune up eta, just like the other methods have to take time for tuning?). It would be nice to see a little more than the ESS, also... it sounds like authors have a ground truth, and authors have MCMC estimates... maybe a log-log plot showing the estimates converging to the truth in the limit as the number of samples grows large?
* Discussion section: here's where you can bring up stuff like "It has been argued [5] that the complete framework is given by a general form of the drift term B(z) = [D(z) + Q(z)]∇H(z) + Γ(z), where H(z) is the Hamiltonian, D(z) is positive definite diffusion matrix and Q(z) is skew-symmetric matrix."  Once the reader has looked at the diffusion that is being studied, the point authors are trying to make will be much more obvious.  As it is, I read that part in the beginning, didn't understand the point, then read the body of the paper, then read the conclusions, didn't understand what "the framework of [5]" meant, had to go back to introduction to look up framework [5].... you get the picture.   I think it would have been easier to just save the whole shebang until the discussion.  In fact, you might spend even a bit more time on it than you already do--- Im assuming you are right that the new move doesn't fit into the framework from [5], but it isn't totally obvious to me since that framework looks like it could include quite a lot.

Regarding the "easy simulations" described above.  Some of the algebra for Theorems 1 and 2 is tricky because of the sphere constraint, and all of this is rather new.  It looked basically right to me, but it would be good to verify with an "easy" simulation.  Say, consider trying to sample a random variable $X \in \mathbb{R}^2$ distributed as a mixture of $\mathcal{N}(0,I)$ and $\mathcal{N}(\mu,I)$ for some $\mu \in \mathbb{R}^2$.  Then use some simulations to showcase/validate the theory.

* Theorem 1, for the deterministic ODE.  It would be nice to simulate this, but as I mentioned above it's not actually clear to me that we can easily run this ODE with enough accuracy to avoid artifacts.  I ran some simulations with leapfrog with a pretty small timestep but they behaved pretty weirdly.  In fact they always converged to an 1d attractor in x space, presumably because of some sort of numerical errors?   Is there an ODE solver that can handle this properly?  Maybe the leapfrog integrator from the ESH paper? If so, worth mentioning the integrator and showing it do a good job. Otherwise, if you can't find one, maybe worth mentioning that users should avoid running MCLMC2024 with small eta because it is too numerically unstable (unless perhaps there's some other way to make an MH adjustment and then maybe we can not worry about the artifacts).
* Theorem 2, for the SDE.  Run the SDE you've proposed (with no bounces).  Check that the empirical distribution of first 10,000,000 steps is close to the true density, and gets closer as you add more steps (e.g., in some MMD distance you can get an explicit formula for).  Maybe also compare to MALA in this setting.  It would set my mind at ease.

Overall, great work!

---

### Official Review · Reviewer_1CGR · 2024-04-22
**SDE extension of an important HMC variant with theoretical results**

**Rating:** 7
**Confidence:** 3

**Review:**

## Summary
This paper proposes an SDE promotion of the microcanonical HMC (MCHMC) by introducing a Wiener process noise to the momentum variable, resulting in microcanonical Langevin Monte Carlo (MCLMC). It is proved that for any noise scale, the stationary distribution of the ODE based MCHMC is a stationary solution of the Fokker-Planck equation for the proposed SDE. The implementation of the idea uses the Euler-Maruyama scheme, which is proved to preserve the stationary distribution for any noise level. Furthermore, the ergodicity is proved, indicating a unique stationary distribution. With smoothness and convexity assumptions, the SDE converges to its stationary distribution exponentially fast. The idea is implemented and compared against NUTS in a modern PPL framework. It is shown that the proposed MCLMC is consistently better on Lattice $\phi^4$ theory and two hierarchical Bayesian models.

My judgement is based on the presentation of the theorems and the experimental results. However, it is beyond my knowledge to check every detail of the proofs.

## Positives
- MCHMC is an important HMC variant which has great potentials in solving difficult geometry common in Bayesian inference and probabilistic programming. Developing an SDE extension of it opens up the possibility of applying the idea to other approximate inference algorithms besides MCMC.
- The theoretical part of the work is complete and substantial. See my summary. It is a quite strong result for a stochastic Langevin process to preserve the desired stationary distribution.

## Negatives
- The manuscript is written for people who know about MCHMC, and will be quite dense for anyone unfamiliar with MCMC theories. For the general audience, more background about MCHMC and how it is compared with HMC should be included. At the same time, some proofs could be pushed to the appendix.
- I do not see a comparison between MCLMC and MCHMC in the experiments. From what I understand, both MCHMC and MCLMC outperforms NUTS, but MCLMC has an $\eta$ to tune. MCHMC may have other considerations, though. I would like to see arguments of practical considerations between the two related approaches.

## Question
- I would also like to see discussions about numerical error in discretization. Theorem 3 states that the distribution is preserved by the Euler-Maruyama scheme. Does it mean that the SDE simulation does not require an Metropolis Hastings correction at all?

---

### Official Review · Reviewer_FZnX · 2024-04-24
**The article tries to produce a microcanonical ensemble based Langevin dynamics without dissipation which whilst maintaining ergodicity can reach a target distribution.**

**Rating:** 9
**Confidence:** 4

**Review:**

# Major
1. This is overall a good quality article and is well-written. The performance of the proposed method is about 10x better in terms of effective sample size and 3x better in terms of computational cost based on the two examples presented, both useful metrics in any sampling scheme. However, the article lacks a discussion/justification of working in a microcanonical setting over a canonical one. Based on [Chetrite and Touchette 2015], in a generalized non-equilibrium setting the microcanonical ensemble is asymptotically equivalent to a canonical ensemble through a change of measure, so then what is the advantage of working in a microcanonical setting when (to the best of my understanding) non-equilibrium canonical ensembles offer a broader and richer setup and myriad routes to reach the target distribution. It would add some value to the work if the authors discuss this point in the paper.

2. Some steps seem to have been skipped in the proof of theorem 1, it would be useful to have them  included.

# Minor
Typo in the proof of Lemma 4, line 3: thought>through

# References
Chetrite, Raphaël, and Hugo Touchette. "Nonequilibrium Markov processes conditioned on large deviations." Annales Henri Poincaré. Vol. 16. Springer Basel, 2015.

---

### Official Review · Reviewer_P8cg · 2024-04-25
**Review of Submission 2**

**Rating:** 7
**Confidence:** 2

**Review:**

Hamiltonian dynamics of particle systems has lead to the construction of two approximate samplers for probability distributions,  Hamiltonian Monte Carlo (HMC) and Langevin Monte Carlo (LMC). In order to sample from a target distribution $\rho(x) \propto \exp[-S(x)]$ it suffices to consider the dynamics of a parameter expanded version $\rho(z) \propto \exp[-H(z)]$, where $H(z)$ is the Hamiltonian and $z=(x,\Pi)$ with $\Pi$ the momentum. The time evolution of this Hamiltonian system is governed by the Liouville equation, a system of ordinary differential equations (ODEs) with $\rho(z)$ as the stationary solution. Solving this system numerically and marginalizing out the momentum $\Pi$ allows us to sample from the desired distribution $\rho(x)$. Langevin dynamics is a stochastic variant based on stochastic differential equations (SDE). The dynamics are governed by the Fokker-Planck equations, a collection of partial differential equations with $\rho(z)$ as the stationary solution (canonical ensemble).

The main motivation for this work it the investigation of an open conjecture based on Ma et al. (2015) which ask the minimum dynamics (ODE / SDE) whose equilibrium solution corresponds to the target density $\rho(z) \propto \exp[-S(z)]$. Their result answers the question under the additional restriction that $\rho(z)$ is the stationary distribution for the system. This solution may be overly restrictive as the original goal of sampling is to take samples from $\rho(x)$.

The main contribution of this work is to show that the recently introduced Microcanonical Hamiltonian Monte Carlo (MCHMC) and Microcanonical Langevin Monte Carlo (MCLMC)  (Robnik et al., 2022) are systems with $\rho(x)$ as the stationary solution; it is not required for $\rho(z)$ to be stationary in these systems. Meaning there exist systems of ODEs and SDEs which for which $\rho(z)$ to be stationary that are not fully described by the systems of Ma et al. (2015). Section 2 derives the stationarity of $\rho(z)$ for MCHMC and section 3 derives stationarity of $\rho(z)$ for MCLMC. Sections 4-6 show that MCLMC is ergodic (uniqueness of stationary solution) and geometrically ergodic -- meaning that it can converge to the stationary solution in finite time. The paper concludes with applications demonstrating MCLMC in Lattice Field Theory and Hierarchical Bayesian models.

If my understanding of the paper is correct, I think this is an nice theoretical contribution to the literature which provides theoretical justification for the recently introduced MCHMC and MCLMC methods and negatively answers an open question from Ma et al. (2015). My main critique of the paper is that it requires a rather large amount of background knowledge, meaning it will be a challenging read for a general audience. Additionally, several notations such $\mathcal{O}$ and $S(x)$ are not formally defined which further hampers the readability.

I was not able to exhaustively check the proofs, particularly section 5 on ergodicity, as my understanding of SDEs is rather rudimentary.

### Comments, Questions, and Suggestions

1. For the HMC/LMC systems that have been proven to be geometrically ergodic, would it be possible to provide a comparison of the rates between these and the one from Theorem 9 for MCLMC?

2. One suggestion for improvement would be to add a table of run time comparisons in numerical study (sec. 7). Intuitively, not having to worry about the momentum should lead to a decrease in the runtime of MCLMC compared to HMC; it would be nice to confirm this.

3. From a readers perspective it would be nice if there was a glossary (or footnotes) of terms translated to machine learning / statistics terminology for some of the physics terminology like canonical ensemble, micro-canonical ensemble, etc.

---

### Meta-Review · Area_Chair_g8hk · 2024-05-24

**Recommendation:** Accept (Poster)
**Confidence:** 4

**Metareview:**

This paper introduces Microcanonical Langevin Monte Carlo (MCLMC), which extends Microcanonical Hamiltonian Monte Carlo (MCHMC) by adding a Wiener process noise to the momentum variable, ensuring ergodicity and exponential convergence to the stationary distribution.

The reviewers agree that the paper provides a solid theoretical contribution, and demonstrates potential for MCLMC in practical applications. There are some concerns about the paper's density and challenging readability, as it requires significant familiarity with prior work.

Overall, the paper seems to be a strong theoretical contribution to the field, and would be an excellent addition to AABI. I encourage the authors to consider how to present this work in the most accessible way for a general AABI audiance.

---

### Decision · Program_Chairs · 2024-05-27

Accept